# Role of Oxidative Stress in Cardiac Dysfunction and Subcellular Defects Due to Ischemia-Reperfusion Injury

**DOI:** 10.3390/biomedicines10071473

**Published:** 2022-06-22

**Authors:** Naranjan S. Dhalla, Anureet K. Shah, Adriana Adameova, Monika Bartekova

**Affiliations:** 1St. Boniface Hospital Albrechtsen Research Centre, Department of Physiology and Pathophysiology, Institute of Cardiovascular Sciences, Max Rady College of Medicine, University of Manitoba, Winnipeg, MB R2H 2A6, Canada; 2School of Kinesiology, Nutrition and Food Science, California State University, Los Angeles, CA 90032, USA; akaur23@calstatela.edu; 3Department of Pharmacology and Toxicology, Faculty of Pharmacy, Comenius University in Bratislava, Odbojarov 10, 832 32 Bratislava, Slovakia; adriana.duris.adameova@uniba.sk; 4Centre of Experimental Medicine, Institute for Heart Research, Slovak Academy of Sciences, Dubravska Cesta 9, 841 04 Bratislava, Slovakia; monika.bartekova@savba.sk

**Keywords:** ischemia-reperfusion injury, oxyradicals and antioxidants, myocardial inflammation, protease activation, intracellular Ca^2+^-overload, cardiac subcellular defects, Ca^2+^-handling abnormalities

## Abstract

Ischemia-reperfusion (I/R) injury is well-known to be associated with impaired cardiac function, massive arrhythmias, marked alterations in cardiac metabolism and irreversible ultrastructural changes in the heart. Two major mechanisms namely oxidative stress and intracellular Ca^2+^-overload are considered to explain I/R-induced injury to the heart. However, it is becoming apparent that oxidative stress is the most critical pathogenic factor because it produces myocardial abnormalities directly or indirectly for the occurrence of cardiac damage. Furthermore, I/R injury has been shown to generate oxidative stress by promoting the formation of different reactive oxygen species due to defects in mitochondrial function and depressions in both endogenous antioxidant levels as well as regulatory antioxidative defense systems. It has also been demonstrated to adversely affect a wide variety of metabolic pathways and targets in cardiomyocytes, various resident structures in myocardial interstitium, as well as circulating neutrophils and leukocytes. These I/R-induced alterations in addition to myocardial inflammation may cause cell death, fibrosis, inflammation, Ca^2+^-handling abnormalities, activation of proteases and phospholipases, as well as subcellular remodeling and depletion of energy stores in the heart. Analysis of results from isolated hearts perfused with or without some antioxidant treatments before subjecting to I/R injury has indicated that cardiac dysfunction is associated with the development of oxidative stress, intracellular Ca^2+^-overload and protease activation. In addition, changes in the sarcolemma and sarcoplasmic reticulum Ca^2+^-handling, mitochondrial oxidative phosphorylation as well as myofibrillar Ca^2+^-ATPase activities in I/R hearts were attenuated by pretreatment with antioxidants. The I/R-induced alterations in cardiac function were simulated upon perfusing the hearts with oxyradical generating system or oxidant. These observations support the view that oxidative stress may be intimately involved in inducing intracellular Ca^2+^-overload, protease activation, subcellular remodeling, and cardiac dysfunction as a consequence of I/R injury to the heart.

## 1. Introduction

Although reperfusion of the ischemic myocardium is beneficial for the improvement of cardiac function, delayed reperfusion is known to cause impaired recovery of contractile activity, induce cardiac arrhythmias, enhance metabolic defects, and produce structural damage to cardiomyocytes in the heart [1,2,3,4,5,6,7]. These abnormalities due to reperfusion of the ischemic heart are termed as ischemia-reperfusion (I/R) injury, which is commonly associated with clinical procedures such as angioplasty, thrombolysis, coronary bypass surgery, and cardiac transplantation. Extensive research over the past four decades has shown that two major mechanisms, namely the development of oxidative stress and the occurrence of intracellular Ca^2+^-overload, as well as myocardial inflammation and alterations in cardiac metabolism, are considered to explain I/R-induced injury to the heart [8,9,10,11,12,13,14,15]. It should also be mentioned that I/R injury not only affects cardiomyocytes and subcellular organelles but is also known to produce dramatic changes in non-cardiomyocyte structures such as vascular smooth muscle, microvasculature, endothelium, fibroblasts, macrophages, mast cells, adrenergic nerve endings, and endogenous renin-angiotensin system, which are present in the myocardial interstitium [5,9,16,17,18]. Furthermore, I/R-induced injury under in vivo conditions is also associated with the activation of neutrophils, leukocytes, platelets, as well as some systemic and central neuro-endocrine systems [7,10,15,19,20,21]. Thus, it can be appreciated that the pathogenesis of I/R-induced injury is of complex nature.

Since oxidative stress, intracellular Ca^2+^-overload, myocardial inflammation and metabolic defects are inter-related mechanisms and affect each other, it is difficult to identify any one of these to be responsible for the induction of I/R-induced injury. However, the involvement of oxidative stress in inducing a wide variety of cardiac abnormalities directly or indirectly during the development of I/R injury seems most prominent [1,5,10,22,23,24]. This view is based on the fact that oxidative stress has been demonstrated to cause Ca^2+^-handling abnormalities, apoptosis, necrosis, fibrosis, autophagy, lipid peroxidation, protein oxidation, irreversible cardiomyocyte damage and arrhythmias [5,21,25,26]. In addition, oxidative stress has been shown to produce activation of different proteases, dramatic changes in cardiac gene expression as well as defects in subcellular organelles such as mitochondria, myofibrils, sarcolemma (SL), and sarcoplasmic reticulum (SR) for inducing cardiac dysfunction due to I/R injury in the heart [5,21,27,28,29]. It is also noteworthy that several antioxidants have been reported to exert beneficial effects in attenuating I/R-induced alterations in cardiac function and other myocardial abnormalities [2,30,31,32]. Some of these I/R-induced changes involving oxidative stress, intracellular Ca^2+^-overload, myocardial inflammation, and cardiac metabolism are depicted schematically in Figure 1. Furthermore, in the present article, we have attempted to summarize the current knowledge regarding the pathophysiology, cardioprotection, and pharmacotherapy of I/R-induced injury to the heart with respect to highlighting its functional significance. Special emphasis has been laid regarding the generation of oxidative stress as well as its implications for inducing molecular and cellular abnormalities during the development of I/R injury to the heart. Particularly, we have outlined the available evidence to show that I/R-induced alterations in the activities of subcellular organelles are not only attenuated by antioxidants but these changes are also simulated upon exposure of the heart to oxidative stress-generating systems.

## 2. I/R-Induced Generation of Oxidative Stress and Its Implication in Heart Disease

Since the status of oxidative stress in the heart is determined by a balance between the formation of reactive oxygen species (ROS) as well as numerous endogenous oxidants and the presence of various antioxidant systems [2,20,21,33,34], it is important to briefly discuss several components of oxidative stress before indicating its involvement in inducing I/R-linked abnormalities. It is pointed out that ROS mainly include superoxide radicals and hydroxyl radicals as well as oxidants such as hydrogen peroxide (H_2_O_2_) and hypochlorous acid (HOCl); their concentrations are markedly increased upon induction of I/R injury. On the other hand, the activities of various endogenous enzymes such as superoxide dismutase, catalase, and glutathione peroxidase, which serve as antioxidant defense mechanisms are depressed in I/R-perfused hearts [35,36]. Furthermore, nuclear factor erythroid-2 related factor 2 (Nrf2) and various microRNAs, which regulate different antioxidant systems, were decreased due to I/R injury [21,37]. In addition, several antioxidants such as ascorbic acid, glutathione, ubiquinol 9, and vitamin E were decreased upon subjecting the heart to conditions of oxidative stress [2,38]. These observations are consistent with the view that I/R injury to the heart is associated with the development of oxidative stress both as a consequence of increased formation of ROS as well as depressions in the level of antioxidant defense systems.

It is noteworthy that mitochondria are the major source of ROS because of the impaired electron transport chain and depressed oxidative phosphorylation activity in I/R hearts [11,39,40]. Although several factors are considered to be responsible for I/R-induced production of ROS in mitochondria, uncoupling of mitochondrial proteins as well as metabolic overloading due to increased fatty acid flux and accumulation of succinate have been shown to be involved in proton leakage from mitochondria [41,42]. Furthermore, increased formation of nitric oxide due to elevated levels of nitric oxide synthase in the endothelium has been reported to result in the production of peroxynitrite and subsequent nitrosative stress at initial stages of I/R injury [43]. It is pointed out that ROS production is also increased due to the activation of several cellular and neuronal systems such as various leukocytes (e.g., neutrophils), the sympathetic nervous system, and the renin-angiotensin system during the development of I/R-induced injury to the heart [20,21]. In this regard, myeloperoxidase released from leukocytes has been reported to play an important role through the formation of microbicidal reactive oxidants [44] whereas the activation of monoamine oxidase has been demonstrated to generate H_2_O_2_ upon oxidative deamination of catecholamines released from the sympathetic nervous system during the development of I/R-induced injury [45,46]. In addition, angiotensin II formed by the activation of renin-angiotensin system due to I/R injury has been shown to promote the generation of ROS as a consequence of the activation of NADPH oxidase (NOX), present in the plasma membrane (NOX2) and intracellular organelles (NOX4) [21,47]. Thus, a wide variety of enzymes as well as cellular and neuronal systems are involved during the development of oxidative stress.

Generation of oxidative stress and nitrosative stress due to I/R injury are not only known to cause cardiac dysfunction, these pathological factors have also been shown to be involved in the formation of pro-inflammatory agents as well as disruption of different signal transduction pathways [20,21,48,49]. Particularly, these pathogenic factors are known to induce myocardial cell damage, apoptosis, necrosis, and fibrosis as well as different defects in cardiac gene expression and myocardial metabolism in hearts subjected to I/R injury. There also occurs the activation and infiltration of polymorphonuclear leukocytes which will promote the development of myocardial infarct and disorganization of several adhesion molecules. Oxidative stress for a prolonged period also induces defects in the endothelial function for the formation of nitric oxide (a vasodilator molecule) and subsequently reduce or block the blood flow to the myocardium. Prolonged oxidative stress in the heart will produce Ca^2+^-handling abnormalities in cardiomyocytes and defects in the SL, SR, mitochondria, and myofibrils mainly as a consequence of the activation of proteases and phospholipases [5,12,20,21,26,50]. A schematic representation of some of the events involved in the generation of oxidative stress in I/R heart and its implications in remodeling of subcellular organelles and cardiac dysfunction are depicted in Figure 2. Depression in SL Na^+^-K^+^ ATPase will produce marked changes in the concentration of Na^+^ and K^+^ in cardiomyocytes which may explain the development of cardiac arrhythmias associated with I/R injury. Furthermore, alterations in SL Na^+^-Ca^2+^ exchange, Ca^2+^-channels and Ca^2+^-gating mechanisms as well as SR Ca^2+^-release and Ca^2+^-pump ATPase due to oxidative stress may account for Ca^2+^-handling abnormalities in cardiomyocytes, and changes in contraction and relaxation processes in I/R hearts. Oxidative stress is also considered to promote the occurrence of mitochondrial Ca^2+^-overload and induce defects in the process of ATP production and release different cytotoxic agents including cytochrome C for the development of I/R-induced injury to the heart. Some events associated with I/R-induced development and consequence of intracellular Ca^2+^-transport systems indirectly through oxidative stress are shown in Figure 3. In addition to inducing intracellular Ca^2+^-overload, oxidative stress has been demonstrated to produce varying degrees of defects in myofibrillar proteins for the loss of Ca^2+^-sensitivity in myofibrils and subsequent depression in cardiac dysfunction due to I/R injury. Thus, oxidative stress can be viewed as the most prominent pathophysiologic factor for the induction of I/R injury to the heart.

## 3. Pathophysiological Aspects of I/R-Induced Injury

The pathophysiology of I/R injury to the heart is of complex nature because it involves the effects of reperfusion which are superimposed upon those of myocardial ischemia [30,51,52,53,54,55,56]. The ischemic insult by occluding the coronary arteries is associated with the lack of oxygen/hypoxia, inability of mitochondria to oxidize substrate, depression of oxidative phosphorylation, and accumulation of hydrogen in cardiomyocytes. The initial events due to myocardial ischemia result in the stimulation of SL Na^+^-H^+^ exchange and SL Na^+^-Ca^2+^ exchange systems as well as elevation in the intracellular concentration of Ca^2+^, depression in energy production, occurrence of some ultrastructural damage, and cessation of contractile activity. These varying degrees of alterations in the heart are dependent upon the duration of myocardial ischemia at early stages. However, if the reperfusion is carried out after a certain period of myocardial ischemia, the changes in cardiac metabolism, Ca^2+^-handling ultrastructure, and contractile function become irreversible and are commonly called as lethal reperfusion injury of the heart. It should be emphasized that I/R injury is not only limited to inducing marked abnormalities in cardiomyocytes but other structures and cells in myocardial interstitium including endothelium and smooth muscle cells of coronary vessels are adversely affected. In addition, I/R injury to the heart is known to produce dramatic changes in macrophages, leukocytes, and platelets, as well as the sympathetic nervous system and the renin-angiotensin system for promoting inflammation and oxidative stress. Thus, I/R injury is considered to result in cellular death (formation of infarct) of the ischemic myocardium by a wide variety of pathological mechanisms.

In addition to oxidative stress and intracellular Ca^2+^-overload, myocardial I/R injury has been shown to release various inflammatory cytokines such as tumor-necrosis factor α (TNF-α), interleukin-1β (IL-1β), and interleukin-6 (IL-6) [57,58,59,60,61]. All these mechanisms are interlinked and are considered to be closely involved in causing I/R-induced myocardial cell damage, apoptosis, necrosis, and fibrosis as well as calcium overload, cardiac arrhythmias, and heart dysfunction. Different regulatory noncoding RNAs including long noncoding RNAs (lncRNA) and microRNAs (miRNAs) have also been reported to play a critical role in the initiation and progression of I/R-induced injury through the expression of target genes for oxidative stress and inflammation [62,63,64]. Thioredoxin-interacting protein has been demonstrated not only to sensitize cardiomyocytes to oxidative stress-induced apoptosis but has also been implicated in the regulation of inflammatory response and glucose homeostasis during the development of I/R injury to the heart [65]. Recently, different arachidonic acid metabolic pathways such as cyclooxygenase pathway, lipoxygenase pathway, and cytochrome P450 monooxygenase pathway have been suggested to be involved in the development of I/R injury [66].

It is noteworthy that I/R injury to the heart not only produces abnormalities in cardiomyocytes for the induction of contractile dysfunction but has also been shown to cause defect in the endothelium for inducing no-flow phenomenon in coronary circulation [67,68,69]. Such a defect in endothelial dysfunction is elicited by oxidative stress and inflammation upon infiltration of leukocytes as well as activation of fibroblasts due to I/R injury [68,69]. Furthermore, upregulation of platelet surface receptors and release of immunomodulatory mediators have been shown to be involved in the modification of endothelial function during the development of myocardial I/R injury [70]. It should be noted that changes in mitochondrial function due to I/R-induced injury not only participate in generating oxidative stress but these organelles are also adversely affected by I/R injury to the heart [71,72]. Particularly there occurs mitochondrial Ca^2+^-overload, which will further depress the energy-producing ability and impair the recovery of cardiac function as a consequence of I/R injury. A combination of both Ca^2+^-overload and oxidative stress is considered to open mitochondrial permeability transition pores involving the participation of both PKC-δ and PKC-ε, and thus releases different proteins for the activation of apoptosis due to I/R injury [73,74]. In addition, I/R-induced injury to the heart has been shown to produce endoplasmic reticulum stress leading to the accumulation of unfolded proteins, and cause Ca^2+^-handling abnormalities due to a marked release of Ca^2+^ as a consequence of SR ryanodine receptor oxidation [75,76]. However, the individual contribution of oxidative stress, inflammation and intracellular Ca^2+^-overload in the genesis of these different myocardial alterations during the development of I/R injury remains to be investigated.

## 4. Cardioprotection in Hearts Subjected to I/R Injury

In order to attenuate the adverse effects of oxidative/nitrosative stress, inflammation and intracellular Ca^2+^-overload in I/R hearts, different redox-based strategies involving endogenous components have been attempted to prevent I/R-induced myocardial cell damage and cardiac dysfunction [77,78,79]. Particularly, interventions such as ischemic preconditioning, ischemic postconditioning, and remote conditioning have been demonstrated to exert cardioprotective actions in improving cardiac performance as well as limiting infarct size and preventing adverse cardiac remodeling due to I/R injury [80,81,82]. It is pointed out that there are several other conditions such as hibernation of myocardium and early stages of myocardial ischemia but their cardioprotective effects against I/R injury have not been examined in details. Although the exact mechanisms for the beneficial effects of these cardioprotective interventions are not fully understood, reductions in the formation of ROS or reactive nitrogen species, levels of lipid peroxidation products and oxidized DNA/RNA bases content as well as activators of redox-based signaling and mitochondrial modulators have been implicated. Modulation of endogenous reducing mechanisms such as thioredoxin and glutathione systems, which are known to scavenge oxyradicals and reduce oxidized proteins through thiol disulfide exchange reactions, is also considered to be involved in cardioprotection [83]. Heme oxygenase-1 protein, which degrades the oxidant heme and generates the antioxidant bilirubin and anti-inflammatory carbon monoxide, has been reported to participate in the intrinsic defense mechanisms for protecting I/R injury [84]. Furthermore, peroxisome proliferator-activated receptor γ (PPARγ), which regulates the gene expression of enzymes involved in glucose and lipid metabolism, was observed as an excellent target for cardioprotection against I/R injury because of its ability to attenuate oxidative stress and inhibit inflammatory response [85]. It is also noteworthy that different gaseous molecules such as nitric oxide, hydrogen sulfide, and hydrogen have been shown to prevent I/R injury to the heart because of their antioxidative, anti-inflammatory, antiproteolytic, and antiapoptotic activities [86,87,88].

Several experimental studies have indicated that cardioprotection by ischemic preconditioning is associated with increases in some enzymes as well as translational and transcriptional factors, which are involved in the regulation of innate detoxifying and antioxidant systems in I/R hearts [89,90,91,92,93]. In this regard, it is pointed out that the elevated level of O-linked β-N-acetylglucosamine (O-GlcNAc), which modifies numerous biological processes post-translationally, has been shown to be involved in reduction of intracellular Ca^2+^-overload, attenuation of mitochondrial permeability transition pore opening, suppression of endoplasmic reticulum stress, and modification of inflammatory response [89]. The ischemic preconditioning-induced activation of hypoxia-inducible factor-1α (HIF-1α), an oxygen sensitive transcription factor, has been reported to improve mitochondrial function, decrease oxidative stress, interact with non-coding RNAs, and activate cardioprotective signaling pathway as well as downstream protective genes [90]. The activation of mitochondrial aldehyde dehydrogenase 2, which detoxify reactive aldehydes, has also been shown to play a central role in cardioprotection because it inhibits opening of the mitochondrial permeability transition pores, attenuates autophagy, and prevents I/R-induced arrhythmias [91,92]. Furthermore, increased levels of redox-sensitive microRNAs, which regulate some components of the cellular antioxidants, interact with proteasomes and modify DNA repair system, have been implicated in cardioprotection of I/R hearts [93]. The activation of Nrf2, a transcription factor that controls the expression of various antioxidant genes, has also been demonstrated to play a pivotal role in enhancing endogenous antioxidant defenses in hearts subjected to I/R injury or myocardial infarction [94,95,96]. It should be mentioned that the induction of stem cells in I/R hearts, where these can differentiate into target tissues and produce trophic paracrine signaling to suppress injury, has been claimed to be of potential therapeutic value [97]. However, it needs to be emphasized that in spite of the wide variety strategies, which have been identified to explain the mechanisms of cardioprotection, a great deal of future research work needs to be carried out to make any meaningful conclusion.

## 5. Pharmacotherapy of I/R Injury to the Heart

In view of the complex pathophysiology of I/R injury involving oxidative stress, inflammation, intracellular Ca^2+^-overload and metabolic defects, various types of pharmacological agents, acting on diverse molecular targets, have been shown to exert beneficial effects in different experimental models of I/R injury [2,4,5,14,20,26,31]. Several antioxidants, Ca^2+^-antagonists, β-adrenoceptor blockers, angiotensin II antagonists, and metabolic modulators have been reported to improve cardiac function, prevent arrhythmias, and attenuate cellular damage in hearts subjected to I/R injury [20,29,30,98,99]. Some phosphodiesterase inhibitors such as pentoxifylline have been shown to prevent I/R-induced cardiac dysfunction by reducing the activation of NF-κB and TNF-α content [100], whereas several TNF-α inhibitors including etanercept exert therapeutic effects by reducing myocardial inflammation and oxidative stress [25,101]. Both leupeptin and compound MDL28170 (inhibitors of matrix metalloproteinases) were observed to prevent I/R injury by depressing the activation of proteolytic enzymes in the heart [26,102,103]. Furthermore, various aldosterone receptor antagonists and sodium-glucose cotransporter 2 (SGLT2) inhibitors have been observed to attenuate I/R injury as well as infarct size due to myocardial infarction by multiple mechanisms including inflammation and oxidative stress [104,105]. Because of the multifactorial basis of I/R injury, a wide variety of drugs such as cyclosporin, colchicine, tocilizumab, glucagon-like peptide 1 antagonist and modulators of different protein kinases (acting at different target sites) have been demonstrated to limit myocardial infarct size as well as prevent cardiac arrhythmias, cellular necrosis, apoptosis, and metabolic defects [99,106,107,108]. In addition, these agents have been shown to promote endothelial and vascular functions, enhance flow, and improve cardiac function.

Since the antioxidant reserve is depressed in hearts exposed to I/R injury for a prolonged period, it is considered appropriate to enhance the endogenous antioxidant systems (either by inducing increases in their activities or by supplementation with exogenous antioxidant) if the adverse effects of I/R injury have to be reversed [29,30,109,110]. In this regard, exercise-induced increases in endogenous antioxidants and nutritional supplements with polyphenolic compounds from foods such as grapes, cocoa, and soy have been reported to limit I/R-induced myocardial cell damage [110]. While several synthetic antioxidants, *N*-acetylcysteine and *N*-mercaptopropionylglycine, have been shown to attenuate I/R injury in animal experimentation, clinical trials have not provided indisputable evidence for favorable action by using these antioxidants in humans [31,32,111]. Untargeted applications of insufficient doses and/or delayed administration following I/R injury may explain the ineffectiveness of these antioxidants in clinical studies [34,112]. On the other hand, a systematic approach for the use of antioxidant vitamins was proposed to offer a novel opportunity to ameliorate the lethal I/R injury [108]. In fact, different studies have revealed that dietary antioxidant vitamins such as vitamin A, C, E, and β-carotene are effective in preventing major cardiovascular events associated with I/R injury [24,113,114]. Some other antioxidants including lazaroid U83836E and liproxstatin-1 have been reported to exert protective effects against I/R injury by targeting protein kinase C and ferroptosis, respectively [115,116]. A non-enzymatic antioxidant selenium, which is an essential component of oxyradical scavengers such as glutathione peroxidase and thioredoxin reductase, has been demonstrated to attenuate I/R injury by regulating the gene expression of these selenoenzymes [117,118]. 

By virtue of their ability to function as the major source for generating oxidative stress during the development of I/R injury, mitochondria are regarded as a main target of several pharmacological agents for improving cardiac function [27,119,120,121,122,123]. Cardioprotection by various therapeutic interventions is achieved by attenuating alterations in different mitochondrial events such as oxidative phosphorylation, mitochondrial membrane potential, Ca^2+^-overload, permeability transition pore formation, leakage of different apoptotic and necrotic factors, mitochondrial cardiolipin content, as well as NAPDH oxidase 2 activity. Alteration of I/R-induced changes in the heart and the left ventricular function by gypenoside, a herbal medicine, was associated with preservation of mitochondrial enzymatic activities of complex 1, II, and IV in the respiratory chain as well as the activity of citrate synthase for energy generation [124]. The beneficial effects of cyclosporine A in I/R injury were found to be due to its action on desensitizing the mitochondrial permeability transition pore opening in the myocardium [125,126]. Likewise, the antioxidant activity of melatonin in reducing the adverse effects of I/R injury was also shown to be related to its inhibitory action on the mitochondrial permeability transition pore opening as well as up-regulation of cytochrome c oxidase activity [127,128]. In addition, nicorandil, a mitochondrial ATP-sensitive potassium channel opener, has been demonstrated to prevent I/R-induced injury to the myocardium, alleviate cardiomyocyte necrosis, attenuate endothelial dysfunction, and improve blood flow as well as cardiac function [129]. These experimental observations indicate that several cardiac alterations induced by I/R injury are prevented by treatment with a wide variety of pharmacological agents including antioxidants; however, extensive research work needs to be carried out to establish if these interventions are able to reverse the I/R-induced myocardial abnormalities. It is desirable to include all doses and administration routes of these drugs for indicating their cardioprotective effects against I/R injury, but these issues require detailed description and thus are not considered within the scope of this article. 

## 6. Evidence for the Role of Oxidative Stress in I/R-Induced Cardiac Dysfunction and Subcellular Defects

Although several experimental studies have revealed that oxidative stress generated during the development of I/R injury is associated with cardiac dysfunction, occurrence of intracellular Ca^2+^-overload, metabolic abnormalities, and subcellular defects for Ca^2+^-handling, the cause-and-effect relationships among these alterations are not fully understood. Accordingly, we have analyzed some of the existing information to provide evidence for the role of oxidative stress in inducing cardiac dysfunction and subcellular defects as a consequence of I/R injury upon perfusing the isolated hearts in the absence and presence of an oxyradical scavenging mixture or antioxidants [130,131,132,133,134,135,136,137,138]. Furthermore, data were also analyzed to examine if I/R-induced adverse effects in the heart are simulated upon perfusing with an oxyradical generating systems or H_2_O_2_, a well-known oxidant. For the purpose of inducing I/R injury, isolated perfused rat hearts were subjected to 30 min of global ischemia followed by different periods of reperfusion whereas the effects of oxidative stress were examined by perfusing the hearts with an oxyradical generating mixture (xanthine plus xanthine oxidase) or H_2_O_2_ for 30 min. The data in Table 1 show that various parameters of the left ventricular (LV) function such as developed pressure, end-diastolic pressure, +dP/dt and -dP/dt were markedly depressed by I/R injury whereas the LV end-diastolic pressure, as well as H_2_O_2_, malondialdehyde and total Ca^2+^ content were increased. These alterations in I/R-induced cardiac function, oxidative stress parameters, and Ca^2+^ content were greatly attenuated by the presence of superoxide dismutase plus catalase in the perfusion medium (Table 1). The data in Table 2 indicate that depressions in both LV-developed pressure and SL Na^+^-K^+^ ATPase activity by I/R injury were associated with increases in the activity of both calpain and matrix metalloproteinase enzymes; these effects of I/R on cardiac function, Na^+^-K^+^ ATPase and proteolytic enzyme activities were markedly attenuated by the presence of antioxidants such as *N*-acetylcysteine and mercaptopropionylglycine. Furthermore, the adverse effects of I/R injury on all these parameters were simulated upon perfusing the heart with xanthine plus xanthine oxidase mixture or H_2_O_2_ (Table 2). It may also be noted that the activities of SL Na^+^-Ca^2+^ exchange and ATP-dependent Ca^2+^-uptake as well as SL Ca^2+^-stimulated ATPase were depressed upon subjecting the heart to I/R injury and these alterations were prevented by the presence of superoxide dismutase plus catalase mixture (Table 3). It may also be seen from Table 3 that SL Na^+^-Ca^2+^ exchange and Ca^2+^-pump activities were depressed upon perfusing the heart with xanthine plus xanthine oxidase and these changes were prevented by the presence of superoxide dismutase plus catalase mixture in the perfusion medium.

The data in Table 4 show that depression of the LV-developed pressure was also associated with decreases in different SR activities such as Ca^2+^-uptake, Ca^2+^-pump ATPase, Ca^2+^-release, and ryanodine binding upon subjecting the heart to I/R injury or perfusion with xanthine with xanthine plus xanthine oxidase as well as H_2_O_2_. It may also be seen from Table 4 that I/R-induced depressions in SR Ca^2+^-pump and Ca^2+^-release activities were prevented by the presence of superoxide dismutase plus catalase in the perfusion medium. Furthermore, data in Table 5 indicate that depressions in both LV-developed pressure and LV end-diastolic pressure by I/R-injury and perfusion with xanthine plus xanthine oxidase or H_2_O_2_ were associated with depressed mitochondrial state 3 respiration and oxidative phosphorylation. These adverse effects of I/R on cardiac function and mitochondrial function were markedly attenuated by the presence of superoxide dismutase plus catalase mixture in the perfusion medium (Table 5). Subjecting the hearts to I/R injury as well as perfusion with xanthine plus xanthine oxidase or H_2_O_2_ also showed depression in LV-developed pressure and myofibrillar Ca^2+^-stimulated ATPase activity (Table 6). These effects of I/R injury were prevented by the presence of an oxyradical scavenger (superoxide dismutase plus catalase mixture) as well as by an antioxidant (*N*-acetylcysteine). Although I/R injury did not affect myofibrillar Mg^2+^-ATPase, the activity of this enzyme was increased upon perfusion with xanthine plus xanthine oxidase as well as H_2_O_2_ (Table 6); the exact reason for the activation of myofibrillar Mg^2+^-ATPase by oxyradical generating system or oxidant is not clear at present. Nonetheless, the overall information described here indicates that there is a linear association between the depression in cardiac performance and changes in subcellular functions related to cation homeostasis, Ca^2+^-handling, energy production and generation of contractile activity during the development of I/R injury. It is noteworthy that I/R-induced alterations in cardiac function and subcellular activities were not only attenuated by oxyradical scavengers or antioxidants but these changes were also simulated upon perfusing the heart with oxyradical generating system or oxidant. These observations provide a compelling evidence that oxidative stress plays a critical role in the pathophysiology of I/R injury.

## 7. Concluding Remarks

From the foregoing discussion, it is evident that there occurs a lack of oxygen, accumulation of intracellular H^+^ due to the inability of mitochondria to oxidize substrates, increase in the concentration of intracellular Ca^2+^ due to the activation of SL Na^+^-H^+^ exchange and Na^+^-Ca^2+^ exchange systems, as well as loss of contractile activity in the ischemic heart. All these alterations are reversible if the reperfusion is carried out during early periods of the ischemic insult but delayed reperfusion has been shown to produce irreversible changes in inflammation and depletion of energy stores in the myocardium. Mitochondrial defects in respiratory chain as well as changes in different enzymes such as NADPH oxidase, nitric oxide synthase, and monoamine oxidase are the major sources of oxyradicals and oxidants in I/R hearts. Excessive formation of reactive oxygen/nitrogen species and depression in the different antioxidant defense systems result in oxidative/nitrosative stress during the development of I/R injury. Furthermore, the occurrence of intracellular Ca^2+^-overload appears to be due to increased membrane permeability as well as changes in SL and SR Ca^2+^-handling systems. On the other hand, I/R-induced myocardial inflammation is a consequence of the release of inflammatory cytokines including TNF-α from macrophages in the cardiac interstitium as well as peripheral neutrophils which enter the injured myocardium. It is difficult to clearly indicate the cause-and-effect of oxidative stress, myocardial inflammation, or intracellular Ca^2+^-overload with I/R injury because adverse effects of these pathological factors are inter-related. In this regard, it is pointed out that both oxidative stress and myocardial inflammation are known to cause subcellular Ca^2+^-handling abnormalities, mitochondrial Ca^2+^-overload, and depression in energy production. Ca^2+^-abnormalities in SL and SR membranes as well as loss of myofibrillar Ca^2+^-sensitivity have also been shown to occur due to the activation of different proteases and modification of cardiac gene expression by both intracellular Ca^2+^-overload and oxidative stress. Furthermore, opening of mitochondrial permeability transition pore and leakage of various mitochondrial cytotoxic components as well as different apoptotic and necrotic factors in the cytoplasm have been reported to occur by oxidative stress in combination with mitochondrial Ca^2+^-overload. Taken together, these observations and other information in the literature suggest that oxidative stress plays a pivotal role in the development of I/R-induced cardiac dysfunction and myocardial cell damage. Some salient events involving oxidative stress, intracellular Ca^2+^-overload, as well as SL, SR, myofibrillar, and mitochondrial defects for the occurrence of cardiac dysfunction due to I/R-induced injury are depicted in Figure 4.

Several pathophysiological studies in different experimental models of I/R injury have identified various major targets in the myocardium for the development of cardioprotective strategies. Some of these entities include metabolic defects, Ca^2+^-handling abnormalities, lipid peroxidation, protease activation, signal transduction for apoptosis and necrosis, transcriptional factors for maintaining redox homeostasis, and mitochondrial K^+^-ATP channels. Although numerous pharmacological agents, including Ca^2+^-antagonists, β-adrenoceptor blockers, angiotensin II antagonists, metabolic modulators, cyclosporin A, and nicorandil have shown to exert beneficial effects in attenuating I/R injury in animal studies, their results in human clinical trials are not conclusive. On the other hand, strategies such as ischemic preconditioning and ischemic postconditioning, which enhance antioxidant levels, have shown a great promise for cardioprotection in both animals and clinical investigations. Likewise, antioxidant vitamins (vitamin A, C, E, and β-carotene), unlike synthetic antioxidants, have been demonstrated to improve cardiac function and reduce myocardial damage. Such observations support the concept that oxidative stress may be intimately involved in the pathophysiology of I/R-induced injury to the heart. This view is further attested by the observations that I/R-induced alterations in cardiac contractile activity, oxidative stress markers, Ca^2+^-handling by SL and SR as well as myofibrillar proteins and mitochondrial function were attenuated by oxyradical scavenging mixture. Furthermore, all these I/R-induced adverse effects in the heart were simulated upon perfusion with an oxyradical generating system or an oxidant. Thus, there is great challenge for directing the future research activities for developing appropriate antioxidant interventions for the prevention and/or therapy of I/R injury to the heart.

## Figures and Tables

**Figure 1 biomedicines-10-01473-f001:**
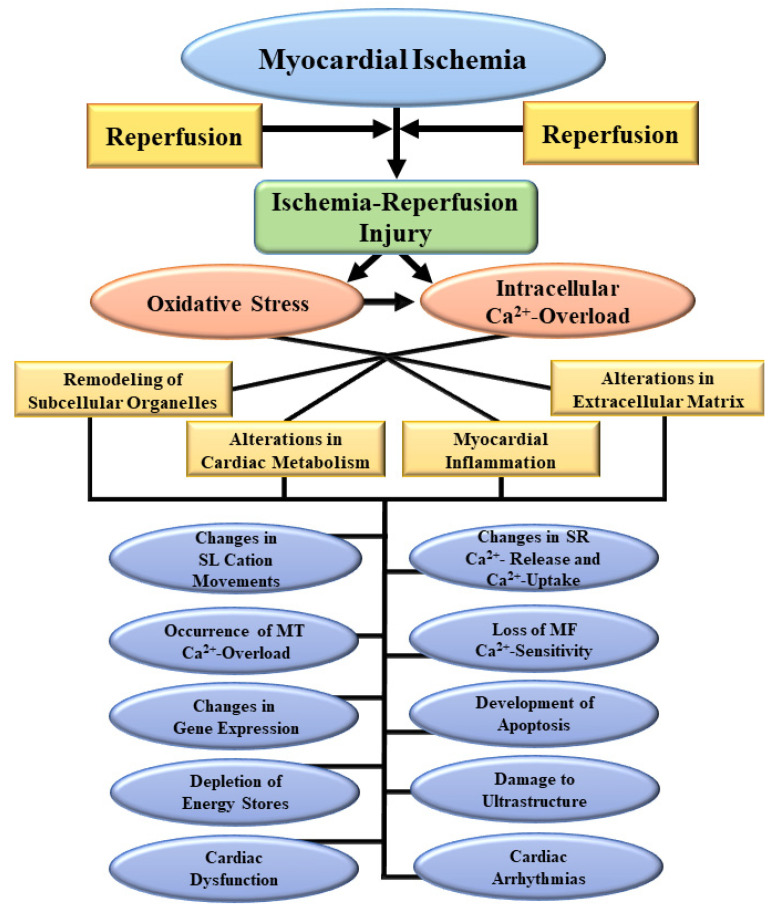
Some myocardial and subcellular abnormalities due to the development of oxidative stress and intracellular Ca^2+^-overload as a consequence of ischemia-reperfusion injury. SL, sarcolemma; SR, sarcoplasmic reticulum; MT, mitochondria; MF, myofibrils.

**Figure 2 biomedicines-10-01473-f002:**
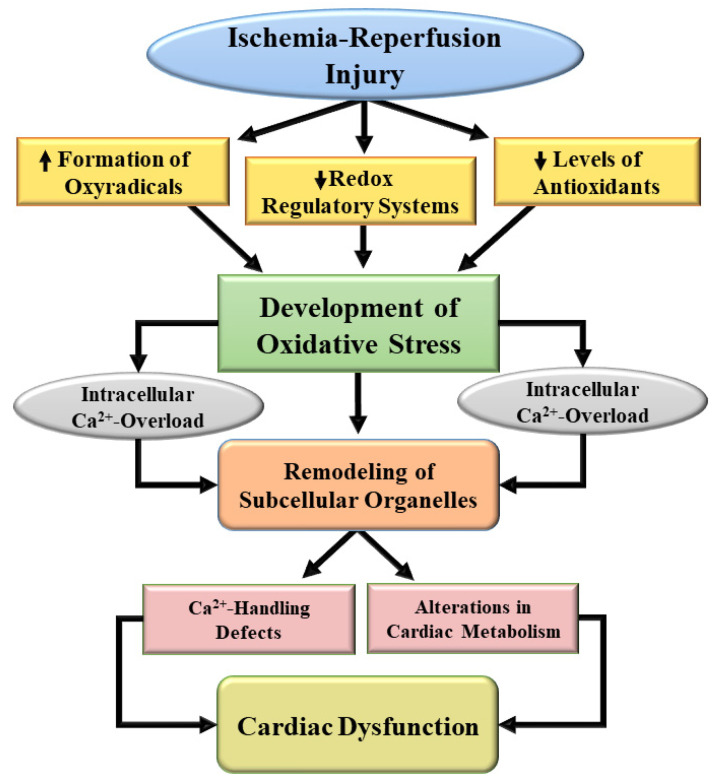
A schematic representation of mechanisms for the development of oxidative stress and subsequent subcellular defects leading to cardiac dysfunction due to ischemia-reperfusion. ↑, increase; ↓, decrease.

**Figure 3 biomedicines-10-01473-f003:**
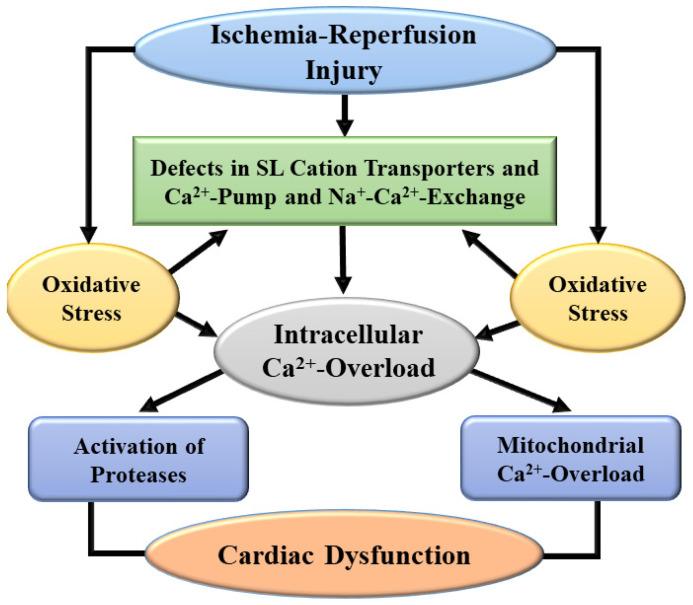
A schematic representation of mechanisms for the development of intracellular Ca^2+^-overload and subsequent subcellular defects leading to cardiac dysfunction due to ischemia-reperfusion injury. SL, sarcolemma.

**Figure 4 biomedicines-10-01473-f004:**
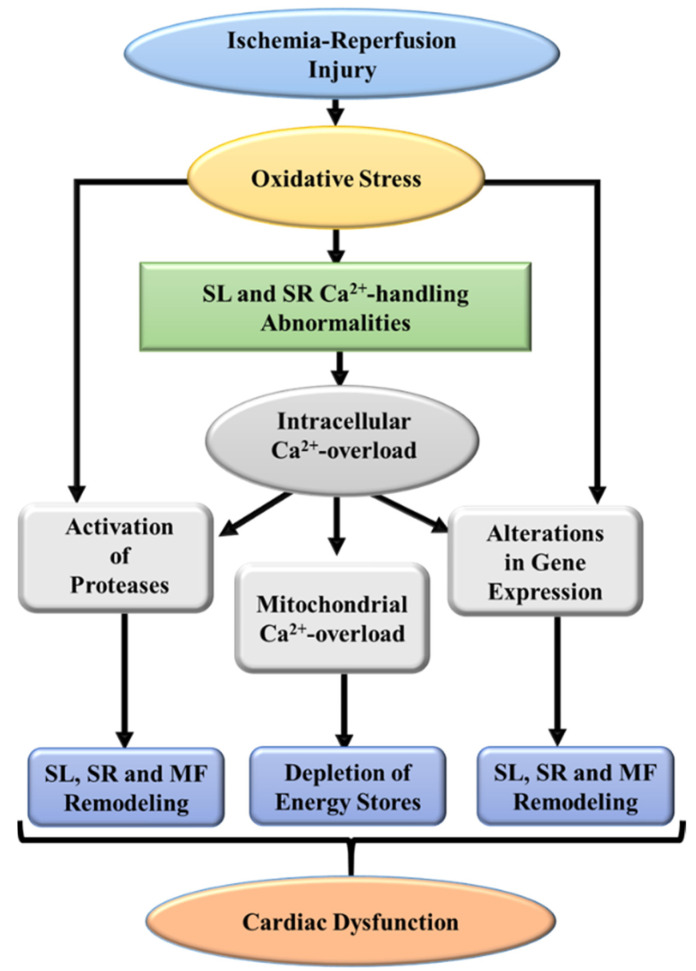
A schematic representation of some major events indicating defects in subcellular organelles leading to cardiac dysfunction due to ischemia-reperfusion injury. SL, sarcolemma; SR, sarcoplasmic reticulum; MF, myofibrils.

**Table 1 biomedicines-10-01473-t001:** Influence of ischemia-reperfusion (I/R) with or without oxyradical scavenger mixture (SOD plus CAT) on cardiac function and myocardial markers for oxidative stress as well as Ca^2+^-content in isolated perfused hearts.

Parameters	Control	I/R	I/R + SOD Plus CAT
A. Cardiac function:			
LV DP (mmHg)	98 ± 3.6	40 ± 2.9 *	72 ± 4.2 †
LV EDP (mmHg)	6.2 ± 0.4	64 ± 4.1 *	36 ± 3.1 †
LV + dP/dt (%)	100 ± 4.2	44 ± 3.1 *	80 ± 3.6 †
LV – dP/dt (%)	100 ± 3.6	35 ± 2.4 *	74 ± 3.0 †
B. Oxidative stress markers:			
H_2_O_2_ content (nmol/g wet wt)	8.4 ± 1.2	38.6 ± 3.9 *	12.3 ± 1.5 †
MDA content (nmol/mg tissue lipids)	3.8 ± 0.6	17.5 ± 3.1 *	5.6 ± 0.8 †
C. Myocardial Ca^2+^:			
Ca^2+^ content (µmol/g dry wt)	8.4 ± 1.2	22.6 ± 2.9 *	9.8 ± 1.6 †

Hearts were subjected to 30 min global ischemia followed by 60 min reperfusion (I/R) in the absence or presence of 80 µg/mL superoxide dismutase (SOD) plus 10 µg/mL catalase (CAT). Control hearts in each experiment were perfused with normal medium for appropriate time. The data are based on the analysis of information in our paper Dhalla et al. [130]. LV—left ventricle; DP—developed pressure; EDP—end diastolic pressure; MDA—malondialdehyde, *—*p* < 0.05 vs. respective control value, †—*p* < 0.05 vs. respective I/R value.

**Table 2 biomedicines-10-01473-t002:** Influence of ischemia-reperfusion (I/R) with or without some antioxidants as well as perfusion with xanthine plus xanthine oxidase (X + XO) or H_2_O_2_ on cardiac function, sarcolemmal Na^+^-K^+^ ATPase activity and protease activities in isolated perfused hearts.

Parameters	Left Ventricle Developed Pressure (mmHg)	Na^+^-K^+^ ATPase Activity ((µmol) Pi/mg/h)	Protease Activities (Relative Fluorescence Units)	
			MMP Activity	Calpain Activity
A. I/R injury/antioxidants:				
Control	119 ± 5.7	28.7 ± 3.8	50 ± 4.3	36 ± 3.1
I/R	44 ± 9.8 *	10.9 ± 3.6 *	525 ± 26.9 *	592 ± 25.9 *
I/R + NAC	114 ± 11.6 †	32.5 ± 3.5 †	163 ± 8.3 †	215 ± 16.5 †
I/R + MGP	121 ± 13.2 †	33.2 ± 4.1 †	152 ± 9.6 †	240 ± 23.7 †
B. Oxidative stress:				
Control	94 ± 7.9	26.9 ± 4.1	56 ± 3.9	40 ± 4.2
X + XO	40 ± 4.2 *	7.6 ± 3.6 *	608 ± 23.8 *	600 ± 15.9 *
H_2_O_2_	55 ± 6.1 *	6.8 ± 2.7 *	450 ± 15.6 *	665 ± 22.7 *

Hearts were subjected to 30 min global ischemia followed by 30 min reperfusion (I/R) in the absence and presence of 100 µM *N*-acetylcysteine (NAC) or 300 µM mercaptopropionylglycine (MGP). Hearts were also perfused for 30 min with 2 mM xanthine (X) plus 60 mU/mL xanthine oxidase (XO) mixture or 100 mM H_2_O_2_ followed by 30 min reperfusion. Control hearts in each experiment were perfused with normal medium for 60 min. The data are based on the analysis of information in our paper Singh et al. [131]. MMP—matrix metalloproteinase, *—*p* < 0.05 vs. respective control value, †—*p* < 0.05 vs. respective I/R value.

**Table 3 biomedicines-10-01473-t003:** Influence of ischemia-reperfusion (I/R) with or without oxyradical scavenger (SOD plus CAT) as well as perfusion with xanthine plus xanthine oxidase (X + XO) or H_2_O_2_ on sarcolemmal Na^+^-Ca^2+^ exchange, Ca^2+^-uptake and Ca^2+^-stimulated ATPase activities in isolated perfused hearts.

Parameters	Na^+^-Ca^2+^ Exchange (nmol Ca^2+^/mg/2 s)	ATP-Dependent Ca^2+^-Uptake (nmol Ca^2+^/mg/5 min)	Ca^2+^-Stimulated ATPase Activity (µmol Pi/mg/h)
A. I/R injury/oxyradical scavenger:			
Control	5.2 ± 0.31	23.4 ± 1.2	11.2 ± 0.7
I/R	3.1 ± 0.29 *	9.7 ± 0.7 *	4.4 ± 0.7 *
I/R + SOD plus CAT	4.7 ± 0.21 †	20.8 ± 1.1 †	9.8 ± 0.6 †
B. Oxidative stress:			
Control	3.8 ± 0.15	24.4 ± 1.0	11.7 ± 1.0
X + XO	1.4 ± 0.20 *	3.6 ± 1.2 *	4.1 ± 0.9 *
X + XO + SOD plus CAT	3.0 ± 0.33 †	22.1 ± 1.4 †	9.2 ± 1.2 †

Hearts were subjected to 30 min global ischemia followed by 5 min reperfusion (I/R) in the absence or presence of 50 U/mL superoxide dismutase (SOD) plus 50 U/mL catalase (CAT). Hearts were also perfused with 2 mM xanthine (X) plus 100 mU/mL xanthine oxidase for 20 min in the absence or presence of SOD plus CAT. Control hearts in each experiment were perfused with normal medium for appropriate period. The data are based on the analysis of information in our papers Dixon et al. [132], Matsubara and Dhalla [133] and Matsubara and Dhalla [134]. *—*p* < 0.05 vs. respective control, †—*p* < 0.05 vs. respective I/R or X + XO group.

**Table 4 biomedicines-10-01473-t004:** Influence of ischemia-reperfusion (I/R) with or without oxyradical scavenger (SOD plus CAT) as well as perfusion with xanthine plus xanthine oxidase (X + XO) or H_2_O_2_ on cardiac function and sarcoplasmic reticular Ca^2+^-uptake and Ca^2+^-release activities in isolated perfused hearts.

Parameters	Left Ventricular Developed Pressure (mm Hg)	Ca^2+^-Uptake Activity (nmol/mg/min)	Ca^2+^-Stimulated ATPase Content (% of Control)	Ca^2+^-Release Activity (nmol/mg/15 s)	Ryanodine Binding (pmol/mg)
A. I/R injury/oxyradical scavenger:					
Control	100 ± 5.2	24.7 ± 1.9	100	9.6 ± 1.5	2.4 ± 0.11
I/R	27 ± 2.8 *	12.5 ± 1.3 *	25 ± 1.9 *	2.8 ± 0.3 *	0.8 ± 0.02 *
I/R + SOD plus CAT	86 ± 4.2 †	22.4 ± 2.8 †	20 ± 2.1	5.3 ± 0.6 †	1.8 ± 0.09 †
B. Oxidative stress:					
Control	100 ± 2.9	28.1 ± 0.7	100	10.1± 1.9	2.3 ± 0.10
X + XO	16 ± 1.8 *	9.3 ± 0.8 *	31 ± 1.4 *	1.5 ± 0.1 *	1.0 ± 0.05 *
H_2_O_2_	27 ± 0.9 *	13.9 ± 1.4 *	58 ± 3.8 *	2.3 ± 0.1 *	0.9 ± 0.04 *

Hearts were subjected to 30 min global ischemia followed by 60 min reperfusion (I/R) in the absence or presence of 50 U/mL superoxide dismutase (SOD) and 75 U/mL catalase. Hearts were also perfused for 20 min with 2 mM xanthine (X) plus 0.03 U/mL xanthine oxidase or 300 µM H_2_O_2_. Control hearts in each experiment were perfused with normal medium for appropriate time period. The data are based on the analysis of information in our paper Temsah et al. [135]. *—*p* < 0.05 vs. respective control value, †—*p* < 0.05 vs. respective I/R value.

**Table 5 biomedicines-10-01473-t005:** Influence of ischemia-reperfusion (I/R) in the absence or presence of oxyradical scavenger mixture (SOD plus CAT) as well as perfusion with oxyradical generating mixture (X plus XO) or H_2_O_2_ on cardiac function and mitochondrial function in isolated perfused hearts.

Parameters	Left Ventricular Developed Pressure (mm Hg)	Left Ventricular end Diastolic Pressure (mm Hg)	State 3 Respiration (ng Atoms O/mg/min)	ADP to O Ratio (nmol ADP/ng atom O)
A. I/R injury/oxyradical scavenger:				
Control	95 ± 7	8.6 ± 0.6	402 ± 12	2.94 ± 0.06
I/R	24 ± 2 *	87 ± 5 *	303 ± 15 *	2.58 ± 0.05 *
I/R + SOD plus CAT	60 ± 2 †	40 ± 4 †	403 ± 21 †	2.80 ± 0.06 †
B. Oxidative stress:				
Control	115 ± 11	10.5 ± 0.7	483 ± 11	2.79 ± 0.07
X + XO	14.6 ± 4.6 *	128 ± 8 *	264 ± 12 *	2.48 ± 0.03 *
H_2_O_2_	28.2 ± 2.3 *	35.7 ± 3.5 *	403 ± 5 *	2.50 ± 0.08 *

Hearts were subjected to 30 min global ischemia followed by 30 min reperfusion (I/R) in the absence or presence of 50 U/mL superoxide dismutase plus 75 U/mL catalase. Hearts were also perfused for 30 min with 2 mM xanthine (X) plus 60 mU/mL oxidase or 100 µM H_2_O_2_. Control hearts in each experiment were perfused with normal medium for appropriate time. The data are based on the analysis of information in our paper Makazan et al. [136]. *—*p* < 0.05 vs. respective control value, †—*p* < 0.05 vs. respective I/R value.

**Table 6 biomedicines-10-01473-t006:** Influence of ischemia-reperfusion (I/R) with or without oxyradical scavenger and antioxidant as well as perfusion with xanthine plus xanthine oxidase (X + XO) or H_2_O_2_ on cardiac functions and myofibrillar ATPase activities in isolated perfused hearts.

Parameters	Left Ventricular Developed Pressure (mm Hg)	Myofibrillar ATPase Activities (µmol Pi/mg/h)	
		Mg^2+^-ATPase	Ca^2+^-Stimulated
A. I/R injury/oxyradical scavenger/antioxidant:			
Control	105 ± 20.3	3.5 ± 0.5	13.3 ± 0.3
I/R	36.4 ± 12.1 *	4.0 ± 0.2	10.7 ± 0.4 *
I/R + SOD plus CAT	71.5 ± 9.5 †	3.1 ± 0.3	12.9 ± 0.2 †
I/R + NAC	117 ± 14.4 †	3.1 ± 0.1	13.9 ± 0.1 †
B. Oxidative stress:			
Control	115 ± 10.1	3.6 ± 0.1	12.7 ± 0.1
X + XO	31 ± 2.8 *	10.7 ± 0.2 *	6.9 ± 0.2 *
H_2_O_2_	––––	5.5 ± 0.2 *	10.9 ± 0.4 *

Hearts were subjected to 30 min global ischemia followed by 30 min reperfusion (I/R) in the absence and presence of 80 µg/mL superoxide dismutase (SOD) plus 10 µg/mL catalase (CAT) or 100 µM *N*-acetylcysteine (NAC). Hearts were also perfused for 30 min with 2 mM xanthine (X) plus 60 mU/mL xanthine oxidase or 100 µM H_2_O_2_ followed by 30 min reperfusion. Control hearts in each experiment were perfused with normal medium for appropriate period. The data are based on the analysis of information in our papers Maddika et al. [137] and Suzuki et al. [138]. *—*p* < 0.05 vs. respective control value, †—*p* < 0.05 vs. respective I/R value.

## Data Availability

Not applicable.

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
