# Peer review of "Role of Oxidative Stress in Cardiac Dysfunction and Subcellular Defects Due to Ischemia-Reperfusion Injury"

_biomedicines, 2022, doi:10.3390/biomedicines10071473_

Round 1

Reviewer 1 Report

The current review tries to summarize the extensive body of literature about ischemia reperfusion injury. This field is extensive but unfortunately some of the major aspects of myocardial ischemia reperfusion injury have not been included and others have been extensively described. (i.e. clearly distinguish between preconditioning, hibernating, ischemia reperfusion injury, staged reperfusion)

The review would benefit from a clear structure and a focus on the major findings in recent years. Please, try to summarize the well known facts in a more concise way and utilize the gained space to describe new findings. There are multiple well received reviews on myocardial ischemia/ reperfusion published already (i.e. Roberto Bolli, Michael Piper and others). 

Try to focus on new findings in the new era of proteomics, single cell sequencing, WGS and other techniques. Please, provide a summary of molecular mechanism and expand your review on potential drug targets. 

Author Response

Reviewer #1

  1. We have gone through the manuscript and have edited the text for making appropriate changes in sentence structure and problems with language.
  2. It is pointed out that some work has been carried out to distinguish between responses of ischemic preconditioning and hibernating myocardium as well as between the effects of reperfusion at different stages of I/R injury, no effort was made to deal with these issues in this review because we wish to indicate other cardioprotective interventions against I/R injury. Nonetheless, a statement (shown in red) regarding this point has now been made in the first paragraph of Section 4.
  3. This review is focused to highlight the role of oxidative stress in inducing changes in subcellular organelles and cardiac function due to I/R injury. Although several excellent reviews on I/R injury to the heart have appeared in the literature (which have been quoted in this article), it is noteworthy that the information regarding pathophysiology and pharmacotherapy of I/R injury as presented in this review is novel and extend our knowledge indicated in other articles. Furthermore, this review presents the comparison of effects of I/R injury and experimentally induced oxidative stress on the functions of various subcellular organelles, which is not provided in other reviews. Thus, these new features of this article make it distinctly different from other articles on I/R injury. Nonetheless, appropriate statements (shown in red) regarding these novel characteristics of this review have been added in the last paragraph of Section 7.
  4. Although we have not used the name of any molecular and biochemical techniques employed for obtaining information regarding various parameters and targets in this article, it is pointed out that we have attempted to provide the latest information on all aspects of the subject matter discussed in this article. This point is readily apparent from the fact that 54 out of 135 references cited in this review are for papers published during 2015-2022. It may be noted that we have analyzed the extensive body of complex literature on the pathophysiology, cardioprotection and pharmacotherapy of I/R injury emphasizing the role of oxidative stress and intracellular Ca2+-overload. Accordingly, as suggested we have now provided a summary of molecular mechanisms and some of affected targets due to I/R injury to the heart with or without drug treatments in the last paragraph of Section 7.

Reviewer 2 Report

The article aim to summarize the current knowledge regarding the pathophysiology, cardioprotection and pharmacotherapy of I/R-induced injury to the heart with respect to highlighting its functional significance. The review is very well designed and clear, being a very complex subject, the authors managed to simplify without losing important information.

The biggest problem is the formatting, it's very unformatted and doesn't show images, even though they are in the text.

General comments:

  1. The introduction is very good, presenting a clear objective.
  2. The item 3. Pathophysiological Aspects of I/R-induced Injury should be summarized, in a picture or on a diagram.
  3. The item 5. Pharmacotherapy of I/R Injury to the Heart needs to be developed further, where the doses used should be added, the routes of administration, as well as the name of the drugs. This information should be summarised in a table.
  4. Table 1 is unformatted and the legend is not complete either.
  5. Line 393 and 394 are unformatted, and I also don't understand what you mean, please clarify.
  6. I don't understand where the values presented in the tables come from, there is no bibliographic references? are they from the author?
  7. The concluding Remarks are good, however I would like to have a more critical summary. I would also like the authors to add the next steps in this area.

Author Response

Reviewer #2

  1. The whole manuscript has undergone computer check for any spelling mistakes.
  2. We have carefully formatted the manuscript and have inserted the images which were missing from the text.
  3. The pathophysiological aspect of I/R injury to the heart emphasizing the role of oxidative stress and intracellular Ca2+-overload are shown in Figures 2 and 3, which were missing in the unformatted manuscript.
  4. The suggestion for a need for giving doses and administration routes of various drugs used for the pharmacotherapy of I/R injury in a Table form is highly appreciated. However, it is pointed out that since these drugs (acting at various targets for modifying different pathophysiological parameters) were given in different animal models under different experimental conditions, it would require extensive description of details and thus we believe it is beyond the scope of this article. Nonetheless, we have added an appropriate statement in this regard in the 3rd paragraph of Section 5. All 6 Tables have now been formatted appropriately and legends as well as bibliographic references for each Table are shown clearly. Please note that the values for all these Tables are based on the analysis of data from our papers.
  5. A critical summary (as shown in red) of the information on molecular mechanisms presented in this article is now given in the last paragraph of the Section 7.

Round 2

Reviewer 1 Report

The new version of the manuscript includes some of the correction suggested by the reviewer. Unfortunately, the overall novelty of the text has not significantly improved and the concluding remarks towards other possible mechanisms of ischemia/reperfusion injury are not sufficient to give a realistic reflection of the literature. The review does not provide significant new insights in its current form.

Author Response

  1. We have made computer checks for any spelling mistakes.
  2. Although we respect the views of this reviewer that this article does not provide significant new insight in its current form, I considered that he/she is entitled to his/her opinion. On the contrary, I carefully outlined the reasons on the last paragraph of the Concluding Remarks (which have now been deleted per the advice of Academic Editor) as this information was already present in the manuscript and I consider this article makes a distinct contribution.

Reviewer 2 Report

The authors have revised the manuscript according to the reviewer's comments.

I recommend accept

Author Response

He had no adverse comments.